Accepted at the ICLR 2024 Workshop on AI4Differential Equations In Science

# Neural Langevin-type Stochastic Differential Equations for Astronomical time series Classification under Irregular Observations

**YongKyung Oh,     Seungsu Kam,     Dong-Young Lim,     &     Sungil Kim**[*]
Ulsan National Institute of Science and Technology, Republic of Korea
`{yongkyungoh, lewki83, dlim, sungil.kim}@unist.ac.kr`

## Abstract

Addressing the classification challenges of irregular time series data in astronomical studies like Large Synoptic Survey Telescope (LSST), this research leverages Neural Stochastic Differential Equations (Neural SDEs) to tackle data irregularity and incompleteness. We analyze a comprehensive analysis to the Neural Langevin-type SDEs' optimal initial condition, which is pivotal role in modelling continuous latent state. Three different strategies for selecting initial condition are compared under regular and irregular scenario using LSST dataset. Our empirical evaluation using Langevin-type SDEs highlights the superiority of static approach over dynamic approaches for initial condition. This discovery highlights the effectiveness of well-chosen initial values of Neural SDEs to enhance the performance of astronomical time series classification under irregular observations.

## 1 Introduction

In the field of observational astronomy, the advent of large-scale surveys like the Large Synoptic Survey Telescope (LSST) marks a transformative era (Allam Jr et al., 2018; Kessler et al., 2019; Muthukrishna et al., 2022; Hložek et al., 2023). These surveys are expected to generate unprecedented volumes of time series data, capturing the subtle and transient events that light up the cosmos. However, the data's inherent irregularity and the presence of observational gaps pose significant challenges for traditional analysis methods. Classifying the diverse array of astronomical phenomena accurately and efficiently from this irregular time series data is not just a necessity but a pivotal step towards unraveling the mysteries of the universe.

The complexity of this task is compounded by the nature of the observational data. Astronomical observations are inherently irregular in time due to factors such as the rotational and orbital dynamics of the Earth, weather conditions, and the operational constraints of the telescopes. Moreover, the presence of missing data or gaps in observation further complicates the data analysis (VanderPlas & Ivezic, 2015; Ivezić et al., 2019; Mitra et al., 2023). Recently, deep learning based method are suggested, but they often assume regularity in data collection and completeness of information, falter in this new and challenging landscape (Chaini & Kumar, 2020; Andrešič et al., 2021; Li et al., 2022).

To address these challenges, this study introduces a novel approach by employing Neural Stochastic Differential Equations (Neural SDEs), a framework that naturally accommodates the stochastic and continuous-time nature of the observational data. In the domain of astrophysics, stochastic perspective of analysis were already discussed (Koen, 2005; Kelly et al., 2014). Compared to the conventional mathematical formalism of stochastic calculus, our method use Neural SDEs, which is inspired by SDEs and combined with neural networks. Neural SDEs not only captures the continuous-time dynamics but also gracefully handles the irregularity and noise inherent in the observational data (Han et al., 2017; Liu et al., 2019; Jia & Benson, 2019; Li et al., 2020). Thus, we expect that Neural SDEs can offer a powerful tool for modeling the intricate dynamics of astronomical phenomena.

A critical aspect of employing the neural differential equations is the selection of initial conditions for the model (Kidger et al., 2020; Morrill et al., 2021; 2022). The choice of initial values significantly

---
[*]Corresponding Author

influences the model's performance, dictating how well the model captures the underlying dynamics of the astronomical events. In this context, we meticulously analyze the impact of different strategies for initializing the Neural SDEs model. Our investigation focuses on three distinct initial value selection methods for the astronomical classification task.

## 2 RELATED WORKS

### 2.1 NEURAL ORDINARY DIFFERENTIAL EQUATIONS (NEURAL ODES)

Consider an input datum $\boldsymbol{x}$ residing in a space of dimension $d_x$, $\boldsymbol{x} \in \mathbb{R}^{d_x}$. Let's define a latent representation $\boldsymbol{z}(t)$ in a $d_z$-dimensional space at any given time $t$, expressed as:

$$\boldsymbol{z}(t) = \boldsymbol{z}(0) + \int_0^t f(s, \boldsymbol{z}(s); \theta_f) \mathrm{d}s, \tag{1}$$

where $\boldsymbol{z}(0) = h(\boldsymbol{x}; \theta_h)$, and $h : \mathbb{R}^{d_x} \to \mathbb{R}^{d_z}$ is an affine transformation parameterized by $\theta_h$, serving as the initializer for $\boldsymbol{z}(t)$. The function $f(t, \boldsymbol{z}(t); \theta_f)$, parameterized by $\theta_f$, is a neural network approximating the derivative $\frac{\mathrm{d}\boldsymbol{z}(t)}{\mathrm{d}t}$. To solve this integral, Neural ODEs employ ODE solvers (e.g. the explicit Euler method, Runge-Kutta method, and so on) (Chen et al., 2018; Rubanova et al., 2019).

### 2.2 NEURAL STOCHASTIC DIFFERENTIAL EQUATIONS (NEURAL SDES)

Neural SDEs extend the concept of Neural ODEs to incorporate stochasticity, describing the random evolution of sample paths as opposed to the deterministic evolution characteristic of ODEs (Tzen & Raginsky, 2019; Look et al., 2020; Williams et al., 2022; Wabina & Silpasuwanchai, 2023). The latent representation in Neural SDEs, $\boldsymbol{z}(t)$, adheres to the following SDE:

$$\boldsymbol{z}(t) = \boldsymbol{z}(0) + \int_0^t f(s, \boldsymbol{z}(s); \theta_f) \mathrm{d}s + \int_0^t g(s, \boldsymbol{z}(s); \theta_g) \mathrm{d}W(s), \tag{2}$$

where $\boldsymbol{z}(0) = h(\boldsymbol{x}; \theta_h)$ and $\{W(t)\}_{t \geq 0}$ signifies a Brownian motion, which is the difference between equation 1. The function $f(\cdot, \cdot; \theta_f)$ acts as the drift function, guiding the systematic, predictable part of the motion. In contrast, $g(\cdot, \cdot; \theta_g)$ serves as the diffusion function, accounting for the random fluctuations in the system, with the latter integral representing the Itô integral. Both the drift and diffusion functions can be effectively modeled using neural networks in Neural SDEs.

## 3 METHODOLOGY

Neural SDEs offer a sophisticated framework for capturing systems characterized by uncertainty and stochasticity, enriching our understanding of complex dynamics. However, achieving stability in Neural SDEs necessitates meticulous design. In this study, we implemented Langevin SDEs (Küchler & Mensch, 1992; Bressloff & Kilpatrick, 2015; Koop et al., 2022) for modelling astrophysical time series. The Langevin SDE is a well-explored topic in stochastic optimization and Markov Chain Monte Carlo (MCMC) algorithms due to its distinctive feature of possessing a unique invariant measure (Gibbs measure). For a more in-depth examination, please refer the following references: Raginsky et al. (2017); Chau et al. (2021); Lim & Sabanis (2021); Lim et al. (2023a;b).

In this study, we follow the formulation of *Neural Langevin-type SDE (Neural LSDE)*, suggested by Oh et al. (2024b) (Please refer the original paper regarding the definition and the proof of stability):

$$\boldsymbol{z}(t) = \boldsymbol{z}(0) + \int_0^t \gamma(\overline{\boldsymbol{z}}(s); \theta_\gamma) \mathrm{d}s + \int_0^t \sigma(s; \theta_\sigma) \mathrm{d}W(s), \tag{3}$$

where $\boldsymbol{z}(0) = h(\boldsymbol{x}; \theta_h)$, and the initial condition plays important role in evolving latent state. $\overline{\boldsymbol{z}}(t) = \zeta(t, \boldsymbol{z}(t), X(t); \theta_\zeta)$ is modified state where $X(t)$ is the controlled path (Kidger et al., 2020), and $\zeta$ is a neural network parameterized by $\theta_\zeta$. This formulation enable model to capture sequential changes (Oh et al., 2024b). Similar with equation 2, the drift term $\gamma(\boldsymbol{z}(t); \theta_\gamma)$ guides the deterministic part of the motion, while the diffusion term $\sigma(t; \theta_\sigma) \mathrm{d}W(t)$ introduces randomness.

The initial condition $\boldsymbol{z}(0)$ can add its variance to the stochastic system and influences the expected value, making the solution more sensitive to the input data. Because of the irregularity and missingness, we consider three different approach to handle initial condition using observation $\boldsymbol{x}$:

(1) **Interpolation method.** Apply natural cubic interpolation using $x$ to make continuous.

(2) **Imputation method.** When there is missing value in $x$, fill mean value instead.

(3) **Static approach.** Ignore partial observations at $t = 0$, and replace value of $x(0)$ with zero.

Here, zero values indicate the average of (partial) observation after normalization. In case of neural differential equations, interpolation methods are widely applied for reflecting the continuous trajectory of latent state. Conventional time series analysis often use the mean imputation for the missingness. Both methods are *dynamic*, because the values are depending on the partial observations $x$. Compare to that, *static* method uses value of zero and initial condition becomes constant at all time.

## 4    EXPERIMENTS

All experiments were performed using a server on Ubuntu 22.04 LTS, equipped with an Intel(R) Xeon(R) Gold 6242 CPU and two NVIDIA A100 40GB GPUs. We followed experimental protocol suggested by Oh et al. (2024b;a) and GitHub Repository[1].

### 4.1    DATASET

The LSST dataset[2] refers to data from the 'Photometric LSST Astronomical Time Series Classification Challenge' (PLAsTiCC)[3] (Kessler et al., 2019), aimed at classifying transient and variable events observed by Large Synoptic Survey Telescope. The challenge[4] involved predicting types of astronomical events based on simulated observations. It included various models of transient and variable sources, realistic observing conditions, and aimed to improve classification methods and study contamination in samples used for dark energy research. Please refer Appendix A for more detailed explanations.

Table 1: Class distribution

| Class | Count | Ratio |
|-------|-------|-------|
| 06 | 69 | 1.4% |
| 15 | 247 | 5.0% |
| 16 | 540 | 11.0% |
| 42 | 763 | 15.5% |
| 52 | 125 | 2.5% |
| 53 | 14 | 0.3% |
| 62 | 306 | 6.2% |
| 64 | 47 | 1.0% |
| 65 | 626 | 12.7% |
| 67 | 136 | 2.8% |
| 88 | 241 | 4.9% |
| 90 | 1554 | 31.6% |
| 92 | 154 | 3.1% |
| 95 | 103 | 2.1% |

Our study utilized preprocessed data as described by Bagnall et al. (2017), comprising 4925 instances, six input dimensions, 36 sequences, and 14 distinct classes for classification purposes. As shown in Table 1, the class distribution is quite imbalanced. Thus we used stratified split technique. The total dataset was partitioned into training, validation, and test sets following a 70:15:15 ratio, respectively.

### 4.2    EXPERIMENTAL PROTOCOL

In the context of modeling continuous latent states for time series data, the initial value $z(0)$ plays a critical role, This initial latent state serves as the starting point from which the model evolves over time. The class of each time series can be identified by using the last value of the latent state, $z(T)$, as the classifier's input. For a detailed implementation of our method, see the Appendix. B.

We evaluated two scenarios: regular and irregular observations. For the irregular observation scenario, we randomly removed 50% of the observations. We normalized all inputs using observed values. Although the distribution between regular and irregular can vary, zero represents the observed mean for both scenarios. We conducted five iterations of cross-validation (CV). We assessed three distinct classification metrics: Accuracy, F1 score, and AUROC (Area Under the Receiver Operating Characteristics) score. Specifically for the AUROC score, we employed the one-vs-rest strategy with micro-averaging, as recommended by Pedregosa et al. (2011).

For the benchmark, we evaluated 20 different methods, which include Recurrent Neural Network (RNN) (Rumelhart et al., 1986; Medsker & Jain, 1999), variations of Long Short-Term Memory (LSTM) (Hochreiter & Schmidhuber, 1997), variations of Gated Recurrent Unit (GRU) (Chung et al., 2014), and a variety of differential equation-based approaches. (See Appendix C for the details.)

---

[1]`https://github.com/yongkyung-oh/torch-ists`
[2]`https://www.timeseriesclassification.com/description.php?Dataset=LSST`
[3]`https://plasticc.org/`
[4]`https://www.kaggle.com/c/PLAsTiCC-2018`

### 4.3 EXPERIMENT RESULTS

Figure 1 shows the example of input data with regular and irregular setting. It is the important to note that the distributions of regular and irregular are different, because of the irregularity and missingness. This distribution shift makes the problem difficult to solve. While Figure 1(b) and (c) shows the varying initial value at $t = 0$, Figure 1(d) use the same initial value for all conditions. The interpolation method is limited in predicting unseen times, rendering the predicted values unreliable. The imputation method relies on each observation, leading to variability across instances. In contrast, the static approach ensures that the initial condition remains consistent across all samples. Based on the three different strategy for the initial condition, state can be changed in the irregular scenario.

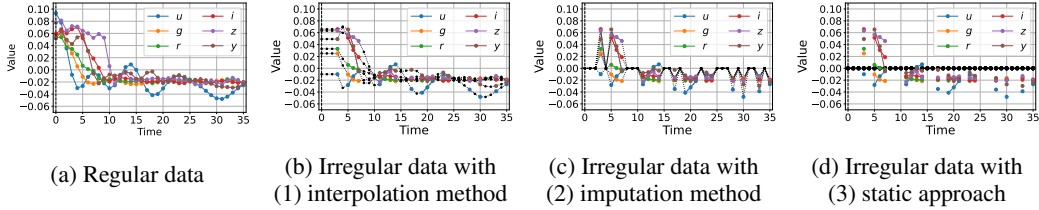

| (a) Regular data | (b) Irregular data with (1) interpolation method | (c) Irregular data with (2) imputation method | (d) Irregular data with (3) static approach |

Figure 1: Regular and irregular observation with six channels and the proposed three approaches (Observed values are only used for learning $z(0)$. Latent state $z \in (0, t]$ is learnt by Neural LSDE.)

Table 2 displays the classification performance in both regular and irregular scenarios. Static approach outperformed the benchmark methods across three metrics. Notably, our method achieved a superior F1 score, a crucial metric for addressing imbalanced classification problems. Furthermore, the use of Neural LSDE with static approach minimized the adverse effects of missing data on performance compared to other methods. It highlights the potential application to real-world astronomical data.

Table 2: Classification performance on regular and irregular setting using LSST data (Average and standard deviation of 5 CV. The **best** and the second best are highlighted)

| Methods | Regular | | | Irregular | | |
| --- | --- | --- | --- | --- | --- | --- |
| | **Accuracy** | **F1 score** | **AUROC** | **Accuracy** | **F1 score** | **AUROC** |
| RNN | $0.428 \pm 0.054$ | $0.218 \pm 0.082$ | $0.882 \pm 0.032$ | $0.344 \pm 0.028$ | $0.101 \pm 0.031$ | $0.819 \pm 0.030$ |
| LSTM | $0.524 \pm 0.057$ | $0.360 \pm 0.057$ | $0.919 \pm 0.017$ | $0.476 \pm 0.024$ | $0.316 \pm 0.048$ | $0.902 \pm 0.010$ |
| BiLSTM | $0.506 \pm 0.032$ | $0.327 \pm 0.055$ | $0.914 \pm 0.008$ | $0.445 \pm 0.029$ | $0.243 \pm 0.036$ | $0.890 \pm 0.008$ |
| PLSTM | $0.457 \pm 0.030$ | $0.273 \pm 0.037$ | $0.898 \pm 0.006$ | $0.426 \pm 0.027$ | $0.264 \pm 0.047$ | $0.876 \pm 0.008$ |
| TLSTM | $0.368 \pm 0.077$ | $0.139 \pm 0.127$ | $0.811 \pm 0.052$ | $0.332 \pm 0.024$ | $0.098 \pm 0.056$ | $0.809 \pm 0.016$ |
| TGLSTM | $0.491 \pm 0.017$ | $0.337 \pm 0.013$ | $0.912 \pm 0.002$ | $0.453 \pm 0.023$ | $0.261 \pm 0.044$ | $0.894 \pm 0.010$ |
| GRU | $0.604 \pm 0.033$ | $0.448 \pm 0.039$ | $0.947 \pm 0.006$ | $0.509 \pm 0.046$ | $0.355 \pm 0.041$ | $0.913 \pm 0.016$ |
| GRU-Simple | $0.354 \pm 0.007$ | $0.157 \pm 0.026$ | $0.824 \pm 0.004$ | $0.329 \pm 0.005$ | $0.086 \pm 0.025$ | $0.809 \pm 0.007$ |
| GRU-$\Delta t$ | $0.540 \pm 0.022$ | $0.305 \pm 0.026$ | $0.927 \pm 0.006$ | $0.520 \pm 0.023$ | $0.300 \pm 0.019$ | $0.921 \pm 0.004$ |
| GRU-D | $0.551 \pm 0.018$ | $0.331 \pm 0.039$ | $0.929 \pm 0.003$ | $0.522 \pm 0.022$ | $0.327 \pm 0.021$ | $0.922 \pm 0.004$ |
| Neural ODE | $0.398 \pm 0.014$ | $0.153 \pm 0.011$ | $0.853 \pm 0.004$ | $0.394 \pm 0.016$ | $0.153 \pm 0.017$ | $0.850 \pm 0.002$ |
| GRU-ODE | $0.436 \pm 0.054$ | $0.230 \pm 0.059$ | $0.887 \pm 0.018$ | $0.434 \pm 0.029$ | $0.232 \pm 0.053$ | $0.887 \pm 0.014$ |
| ODE-RNN | $0.576 \pm 0.021$ | $0.381 \pm 0.043$ | $0.940 \pm 0.005$ | $0.542 \pm 0.015$ | $0.364 \pm 0.023$ | $0.929 \pm 0.003$ |
| ODE-LSTM | $0.412 \pm 0.065$ | $0.235 \pm 0.107$ | $0.850 \pm 0.071$ | $0.373 \pm 0.059$ | $0.164 \pm 0.072$ | $0.822 \pm 0.053$ |
| Neural CDE | $0.381 \pm 0.009$ | $0.161 \pm 0.022$ | $0.849 \pm 0.003$ | $0.372 \pm 0.007$ | $0.141 \pm 0.022$ | $0.845 \pm 0.004$ |
| Neural RDE | $0.317 \pm 0.002$ | $0.041 \pm 0.011$ | $0.796 \pm 0.006$ | $0.316 \pm 0.001$ | $0.037 \pm 0.006$ | $0.794 \pm 0.003$ |
| Neural SDE | $0.396 \pm 0.016$ | $0.210 \pm 0.037$ | $0.862 \pm 0.005$ | $0.390 \pm 0.009$ | $0.175 \pm 0.010$ | $0.856 \pm 0.004$ |
| **Neural LSDE (1)** | $0.402 \pm 0.019$ | $0.186 \pm 0.019$ | $0.866 \pm 0.008$ | $0.398 \pm 0.031$ | $0.183 \pm 0.030$ | $0.860 \pm 0.009$ |
| **Neural LSDE (2)** | $\underline{0.691 \pm 0.012}$ | $\underline{0.556 \pm 0.027}$ | $\underline{0.963 \pm 0.002}$ | $\underline{0.638 \pm 0.009}$ | $\underline{0.511 \pm 0.018}$ | $\underline{0.953 \pm 0.002}$ |
| **Neural LSDE (3)** | $\mathbf{0.695 \pm 0.009}$ | $\mathbf{0.573 \pm 0.041}$ | $\mathbf{0.966 \pm 0.001}$ | $\mathbf{0.648 \pm 0.020}$ | $\mathbf{0.522 \pm 0.027}$ | $\mathbf{0.956 \pm 0.002}$ |

Figure 2 displays the Receiver Operating Characteristic (ROC) curves for the naïve Neural SDE alongside the three distinct strategies. As discussed, the Langevin-type SDE emerges as a promising approach for classifying astronomical time series. Compared to dynamic approaches, the static initial condition enhances performance remarkably. We included the ablation study of the proposed method regarding $\overline{z}(t)$ and the complexity of diffusion term in Appendix B. We found that the integration of the controlled path is crucial for our method, while we reduce the information variance by static initialization. Furthermore, ROC curves for different methods are included in Appendix C.

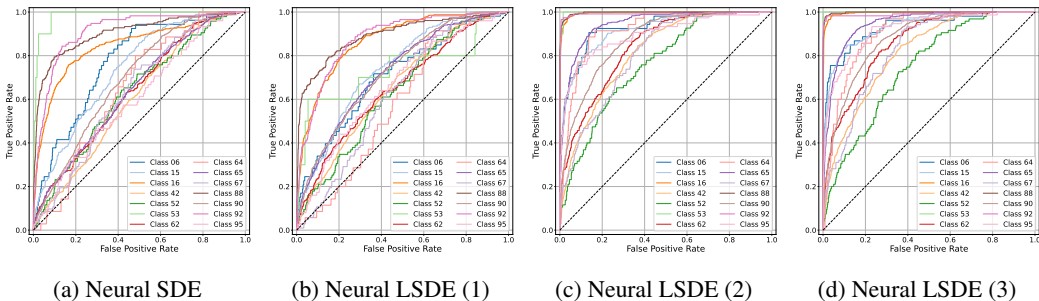

| (a) Neural SDE | (b) Neural LSDE (1) | (c) Neural LSDE (2) | (d) Neural LSDE (3) |

Figure 2: Receiver operating characteristic curves for each class, under the irregular scenario

## 4.4 Ablation Study

The proposed approach comprises several distinct components, including three strategies for the initial condition $z(0)$, the incorporation of a sequential embedding layer $\zeta$ utilizing a controlled path $X$, and the network complexity for the drift and diffusion terms. We used 'Hermite cubic splines with backward differences' (Morrill et al., 2022) for constructing $X$. Regarding the drift network $\gamma$, both the benchmark and proposed methods employ a nonlinear fully-connected layer with `ReLU` activation. We consider the complexity of diffusion term $\gamma$, using Linear affine layer ('L') and Non-linear fully-connected layer with `ReLU` activation ('N'). Table 3 summarizes the considered settings of the ablation study.

Table 3: Ablation study of the model components in the proposed method

| Methods | | | Regular | | | Irregular | | |
|---|---|---|---|---|---|---|---|---|
| $z(0)$ | $\zeta$ | $\sigma$ | Accuracy | F1 score | AUROC | Accuracy | F1 score | AUROC |
| (1) | X | L | $0.403 \pm 0.006$ | $0.152 \pm 0.018$ | $0.861 \pm 0.003$ | $0.396 \pm 0.004$ | $0.152 \pm 0.008$ | $0.855 \pm 0.006$ |
| | | N | $0.390 \pm 0.015$ | $0.172 \pm 0.013$ | $0.859 \pm 0.003$ | $0.391 \pm 0.010$ | $0.166 \pm 0.008$ | $0.852 \pm 0.004$ |
| | O | L | $0.412 \pm 0.017$ | $0.193 \pm 0.027$ | $0.868 \pm 0.007$ | $0.401 \pm 0.019$ | $0.172 \pm 0.020$ | $0.864 \pm 0.006$ |
| | | N | $0.402 \pm 0.019$ | $0.186 \pm 0.019$ | $0.866 \pm 0.008$ | $0.398 \pm 0.031$ | $0.183 \pm 0.030$ | $0.860 \pm 0.009$ |
| (2) | X | L | $0.414 \pm 0.015$ | $0.212 \pm 0.021$ | $0.866 \pm 0.006$ | $0.349 \pm 0.005$ | $0.128 \pm 0.032$ | $0.822 \pm 0.006$ |
| | | N | $0.428 \pm 0.019$ | $0.237 \pm 0.052$ | $0.869 \pm 0.007$ | $0.353 \pm 0.018$ | $0.136 \pm 0.039$ | $0.819 \pm 0.014$ |
| | O | L | $0.666 \pm 0.015$ | $0.534 \pm 0.040$ | $0.961 \pm 0.002$ | $0.638 \pm 0.015$ | $0.509 \pm 0.030$ | $0.954 \pm 0.001$ |
| | | N | $0.691 \pm 0.012$ | $0.556 \pm 0.027$ | $0.963 \pm 0.002$ | $0.638 \pm 0.009$ | $0.511 \pm 0.018$ | $0.953 \pm 0.002$ |
| (3) | X | L | $0.315 \pm 0.000$ | $0.034 \pm 0.000$ | $0.791 \pm 0.003$ | $0.315 \pm 0.000$ | $0.034 \pm 0.000$ | $0.790 \pm 0.002$ |
| | | N | $0.315 \pm 0.000$ | $0.034 \pm 0.000$ | $0.789 \pm 0.002$ | $0.315 \pm 0.000$ | $0.034 \pm 0.000$ | $0.790 \pm 0.001$ |
| | O | L | $0.685 \pm 0.006$ | $0.564 \pm 0.018$ | $0.964 \pm 0.002$ | $0.640 \pm 0.012$ | $0.524 \pm 0.011$ | $0.954 \pm 0.002$ |
| | | N | $0.695 \pm 0.009$ | $0.573 \pm 0.041$ | $0.966 \pm 0.001$ | $0.648 \pm 0.020$ | $0.522 \pm 0.027$ | $0.956 \pm 0.002$ |

Conventional methods typically utilize (1) interpolation and (2) imputation techniques. In contrast, our method applying the (3) static approach disregards partial observations at $t = 0$ and instead initializes all values to zero. This assumes that the time-evolving latent space begins at the average state for all instances. The efficiency of the proposed method is particularly notable when paired with the embedding layer $\zeta$. By neglecting the observation $x$, there's a risk of losing pertinent sample information. Therefore, integrating a controlled path is essential for our method. On the other hand, The network complexity of the diffusion term has a marginal effect on enhancing performance.

## 5 Conclusion

This study navigated the complicated landscape of astronomical time series data, using the power of Neural LSDEs to address challenges of irregularity. The study focused on evaluating initial condition strategies for the Langevin-type SDEs, revealing that static approach outperform dynamic methods in improving data classification accuracy in the presence of noise and irregularities. These findings not only improve the application of Neural SDEs in astronomy, but they also advance time series analysis in general. Furthermore, this research can assist astronomical discovery, fostering sophisticated tools and insights that merge scientific knowledge and data analysis.

ACKNOWLEDGMENTS

This work was partly supported by the Korea Health Technology R&D Project through the Korea Health Industry Development Institute (KHIDI), funded by the Ministry of Health and Welfare, Republic of Korea (Grant number: HI19C1095), the National Research Foundation of Korea (NRF) grant funded by the Korea government (MSIT)(No.RS-2023-00253002), the National Research Foundation of Korea (NRF) grant funded by the Korea government (MSIT)(No.RS-2023-00218913), and the Institute of Information & communications Technology Planning & Evaluation (IITP) grant funded by the Korea government (MSIT) (No.2020-0-01336, Artificial Intelligence Graduate School Program (UNIST)).

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

# A  EXPLANATION OF DATASET

Table 4 provides a summary of the target classes within the LSST dataset in PLAsTiCC. Astronomical observations, that are measured in Modified Julian Date (MJD), are not evenly spaced due to various factors such as the rotation and orbit of the Earth, weather conditions, and operational constraints of the telescope. Also, gaps in the data can occur due to several reasons, such as the object being below the detection threshold of the telescope, the targeted region not being in the field of view, or data being lost or corrupted. These characteristics make the classification problem challenging.

In this dataset, input data consist with six specific filter that allows astronomers to observe light at different wavelengths, denoted as *ugrizy*, such that:

- *u* (Ultraviolet);
- *g* (Green, towards the blue end of the spectrum);
- *r* (Red, mid-spectrum);
- *i* (Near Infrared);
- *z* (Infrared, further along the spectrum than `i`);
- *y* (Infrared, longer wavelength than `z`).

These filters are used to capture and record the brightness of astronomical objects at different wavelengths. By analyzing the intensity of light in each of these filters, astronomers can gain insight into various characteristics of celestial objects, such as their temperature, age, distance, and chemical composition.

Table 4: Class information of the LSST dataset)

| Label | Name | Astrophysical Meaning |
|-------|------|------------------------|
| 06 | μLens-Single | Microlensing events, where the gravitational field of a single object magnifies the light of a background source. |
| 15 | TDE | Tidal Disruption Event, when a star is torn apart by the tidal forces of a supermassive black hole. |
| 16 | EB | Eclipsing Binary stars, systems where the stars pass in front of each other from the observer's perspective. |
| 42 | SNII | Core-collapse Type II Supernova, resulting from the gravitational collapse of a massive star's core. |
| 52 | SNIax | Peculiar Type Iax supernova, similar to Type Ia but fainter and with lower ejection velocities. |
| 53 | Mira | Mira variable stars, a class of red giant stars that pulsate and change brightness in a regular cycle. |
| 62 | SNIbc | Core-collapse Type Ibc Supernova, that have lost their outer hydrogen and possibly helium layers. |
| 64 | KN | Kilonova, emission from the merger of two neutron stars, rich in heavy elements. |
| 65 | M-dwarf | M-dwarf stellar flares, sudden brightness increases due to magnetic activity on low-mass stars. |
| 67 | SNIa-91bg | Peculiar type Ia supernova, 91bg-like events, under-luminous and red compared to typical SNIa. |
| 88 | AGN | Active Galactic Nuclei, the bright and energetic central regions of galaxies, powered by a supermassive black hole. |
| 90 | SNIa | White Dwarf (WD) detonation Type Ia Supernova, a thermonuclear explosion of a white dwarf in a binary system. |
| 92 | RRL | RR Lyrae variable stars, pulsating horizontal branch stars known for their period-luminosity relationship. |
| 95 | SLSN-I | Superluminous Supernovae, specifically Type I, much more luminous than typical supernovae. |

# B  IMPLEMENTATION OF THE PROPOSED METHOD

## B.1  DATA PREPROCESSING

Normalization is a crucial step in preparing data for analysis, ensuring that the inputs have a standard format and scale in each channel (or dimension). Formally, this can be represented as follows:

$$\boldsymbol{x}_{i,\text{normalized}} = \frac{\boldsymbol{x}_i - \mu_i}{\sigma_i}, \quad where \ i \in \{1, 2, \cdots d_x\}.$$

$\boldsymbol{x}_i$ is the original input value (with irregular observation and missing gaps) of $i$-th channel, $\mu_i$ is the mean of the observed values, $\sigma_i$ is the standard deviation of the observed values. Since both regular and irregular distributions are centered around zero (indicating the observed mean), the formula adjusts all input values in relation to this mean, effectively standardizing the distribution of the inputs.

## B.2  LEARNING STRATEGY

The formulation $\boldsymbol{z}(0) = h(\boldsymbol{x}; \theta_h)$ indicates that the initial state is a function of the input data $\boldsymbol{x}$, parameterized by $\theta_h$. Specifically, $\tilde{\boldsymbol{x}} \xrightarrow{\text{mapping } h} \boldsymbol{z}_0 \xrightarrow{t=0} \boldsymbol{z}(0)$, where $\tilde{\boldsymbol{x}}$ is the transformed value with three different initialization strategies, and $\boldsymbol{z}_0$, which is in $\mathbb{R}^{d_z \times T}$ is the mapped value from $\tilde{\boldsymbol{x}}$.

The classification of each time series is determined by the last value of the latent state, denoted as $z(T)$, where $T$ is the final time step. This value is used as an input to a classifier. Mathematically, the classifier can be represented as a function *MLP* that maps the latent state to a class label:

$$\hat{y} = MLP(z(T); \theta_{MLP}),$$

where $\hat{y}$ is the predicted label for given data $(x, y)$. Classifier *MLP* is the two-layer fully-connected Multi-Layer Perceptron (MLP) with `ReLU` activation function.

In our model, we implemented the hyperbolic tangent function as the concluding operation for drift, diffusion, and all additional vector fields, as suggested by Kidger et al. (2020). `tanh` mitigates potential complications arising from exceedingly high values or gradients, which could hinder the learning process and convergence of the model. Furthermore, we used layer-specific learning rates for the final layer of the model (for instance, applying a factor of $\times 100$), promoting a more nuanced and adaptive learning strategy tailored to the classification task at hand.

### B.3 Comparison of learning stability

Figure 3 illustrates the training, validation, and test losses observed when employing the methods under consideration. In comparison to the naïve Neural SDE approach, the Neural LSDE method demonstrates superior performance. Given the challenging nature of astronomical classification problem, neither the validation nor the test losses fully converge to small value. Especially, the Neural LSDE with interpolation (Neural LSDE (1)) exhibits overfitting when contrasted with Neural LSDE with imputation (Neural LSDE (2)) or static approach (Neural LSDE (3)). Consequently, an early-stopping strategy was implemented to enhance classification performance. All models were trained via 100 epochs, but it is terminated when the loss is not improved for 10 consecutive epochs.

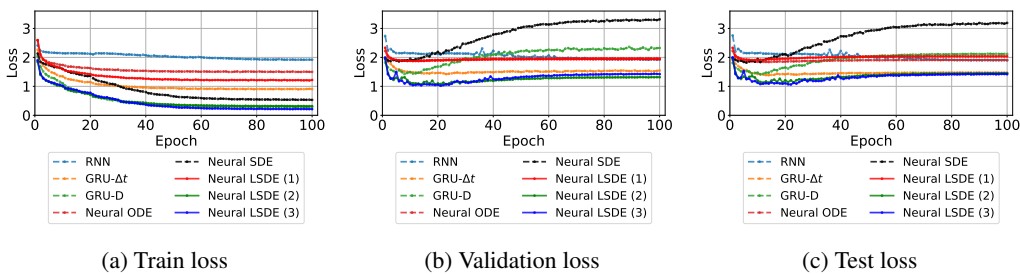

(a) Train loss  (b) Validation loss  (c) Test loss

Figure 3: Comparing stability of loss with irregular setting using the selected methods (Training 100 epochs without early-stopping.)

### B.4 Empirical advantages of the proposed method

**Reduced Sensitivity to Initial Variance:** The system's evolution starts from a consistent, neutral baseline, making the Neural LSDE's behavior primarily dependent on the dynamics defined by $\gamma$ and $\sigma$, rather than the potentially noisy or incomplete initial data $x$. By avoiding the direct influence of $x$ at $t = 0$, the static method mitigates the impact of any missing or erroneous data at the start, which is crucial in fields like astrophysics, where exact initial conditions might not be reliably observable.

**Focus on Learning Dynamic Patterns:** Neural LSDE is controlled by continuous trajectory $z$ using $\zeta$. Thus, Neural LSDE must learn to accurately interpret the dynamics from $\gamma$ and $\sigma$ based purely on data observed during the interval $(0, t]$. Therefore, the trajectory of $z(t)$ is shaped by the model's dynamics and the stochastic nature of the process, rather than initial conditions $x$.

## C Detailed experimental settings and results

All experiments were performed using a server on Ubuntu 22.04 LTS, equipped with an Intel(R) Xeon(R) Gold 6242 CPU and two NVIDIA A100 40GB GPUs. We followed experimental protocol suggested by Oh et al. (2024b;a) and GitHub Repository[5].

---

[5]`https://github.com/yongkyung-oh/torch-ists`

Additionally, we used Python library `torchsde`[6] (Li et al., 2020; Kidger et al., 2021) to formulate and solve the Langevin-type SDEs and Python library `torchcde`[7] (Kidger et al., 2020; Morrill et al., 2021) for the interpolation and the controlled path.

## C.1 BENCHMARK METHODS

For the benchmark, we evaluated 20 different methods, which include Recurrent Neural Network (RNN) (Rumelhart et al., 1986; Medsker & Jain, 1999), variations of Long Short-Term Memory (LSTM) (Hochreiter & Schmidhuber, 1997), variations of Gated Recurrent Unit (GRU) (Chung et al., 2014), and a variety of differential equation-based approaches.

- Conventional recurrent neural network (RNN) (Rumelhart et al., 1986; Medsker & Jain, 1999) is implemented with mean imputation.
- Variations of Long Short-Term Memory (LSTM): LSTM (Hochreiter & Schmidhuber, 1997), Bi-directional LSTM (BiLSTM) (Nguyen et al., 2017), Phased-LSTM (PLSTM) (Neil et al., 2016), Time-aware LSTM (TLSTM) (Baytas et al., 2017), and Time-Gated LSTM (TGLSTM) (Sahin & Kozat, 2018).
- Variations of Gated Recurrent Unit (GRU): GRU (Chung et al., 2014), GRU-$\Delta t$ (Choi et al., 2016), GRU-Simple (Che et al., 2018), and GRU-D (Che et al., 2018)
- Variations of Neural Ordinary Differential Equations (Neural ODEs): Neural ODEs (Chen et al., 2018), GRU-ODE (De Brouwer et al., 2019), ODE-RNN (Rubanova et al., 2019), and ODE-LSTM (Lechner & Hasani, 2020).
- Variations of Neural Controlled Differential Equations (Neural CDEs): Neural CDE (Kidger et al., 2020) and Neural Rough Differential Equation (Neural RDE) (Morrill et al., 2021).
- Variations of Neural Stochastic Differential Equations (Neural SDEs): Neural SDE (Oh et al., 2024b) and Neural LSDE (Oh et al., 2024b).

The (explicit) Euler method was consistently used to solve Neural ODEs, Neural CDEs, and Neural SDEs. Otherwise, all parameters are optimized by Adam optimizer (Kingma & Ba, 2014).

To maintain consistency in comparative analysis, this study employed the same original architecture for all evaluated methods. Nevertheless, recognizing that optimal hyperparameters can differ across methods, the Python library `ray`[8] (Moritz et al., 2018; Liaw et al., 2018) was utilized. This library streamlines the process by automating the selection of hyperparameters aimed at minimizing the validation loss, a notable advancement from prior studies that relied on manual tuning.

Hyperparameter optimization was methodically conducted as follows. The batch size was 128, and the learning rate was varied between $10^{-4}$ and $10^{-1}$, determined through a log-uniform search. The number of layers was chosen from the set $\{1, 2, 3, 4\}$ using a grid search approach, and the hidden vector dimensions were selected from the set $\{16, 32, 64, 128\}$, also via grid search. Optimal hyperparameters are chosen by minimizing the validation loss in the regular scenario. Then, the same hyperparamters are used for both regular and irregular scenarios.

## C.2 RECEIVER OPERATING CHARACTERISTIC CURVES

In Figures 4 and 5, we depict the Receiver Operating Characteristic (ROC) curves for 20 methods under regular and irregular scenarios, respectively. The presence of irregular data notably impacts performance, particularly for conventional methods like RNN. Typically, methods based on differential equations outperform those based on architectural modifications (such as variations of LSTM and GRU). Notably, the proposed Langevin-type SDE method with static method demonstrates the most superior performance. Therefore, these results confirm that initial condition is critical to train the neural differential equation methods, including Neural ODEs, Neural CDEs, and Neural SDEs.

---

[6] https://github.com/google-research/torchsde
[7] https://github.com/patrick-kidger/torchcde
[8] https://github.com/ray-project/ray

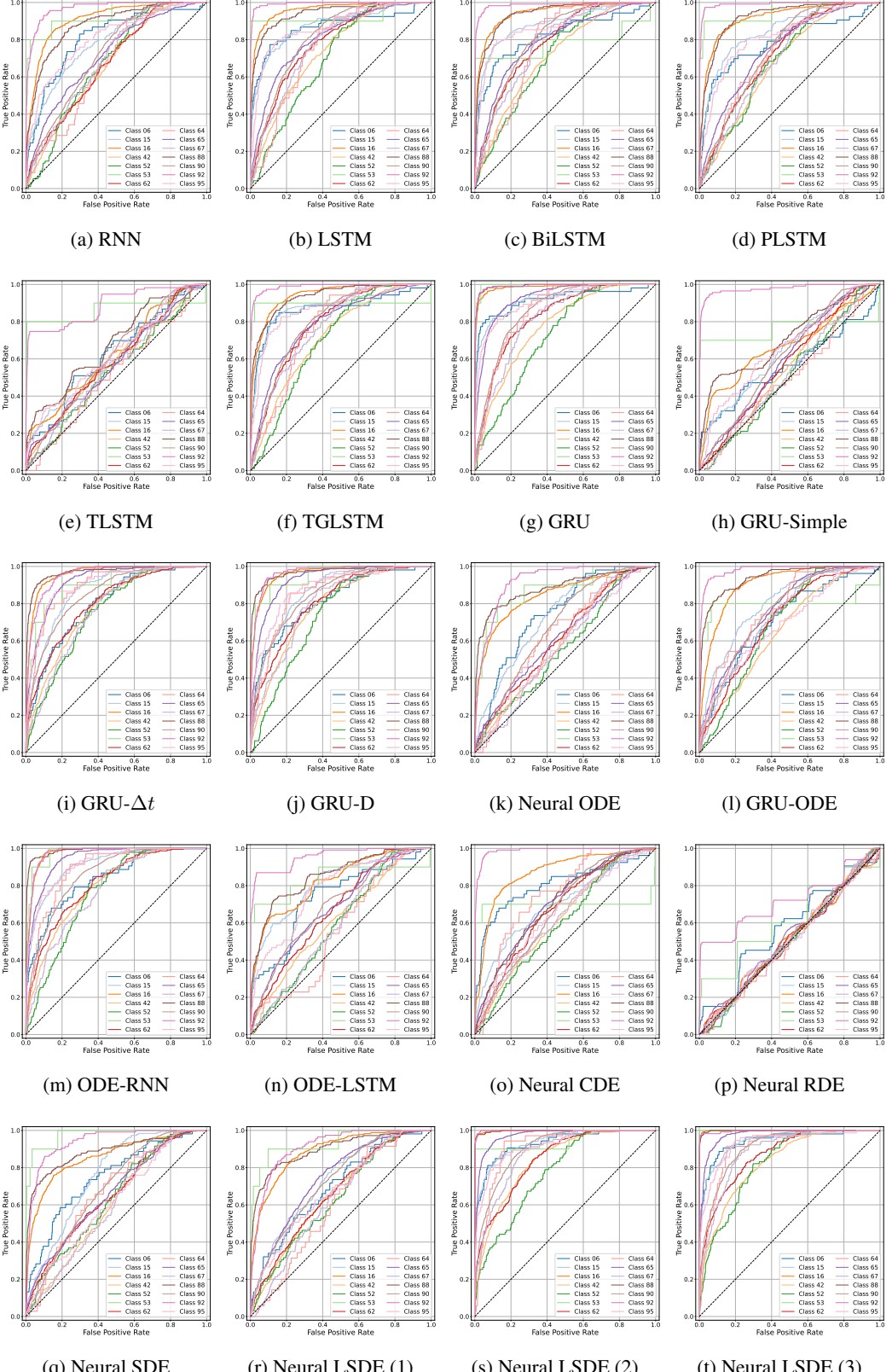

Figure 4: Receiver operating characteristic curves for each class, under the regular scenario

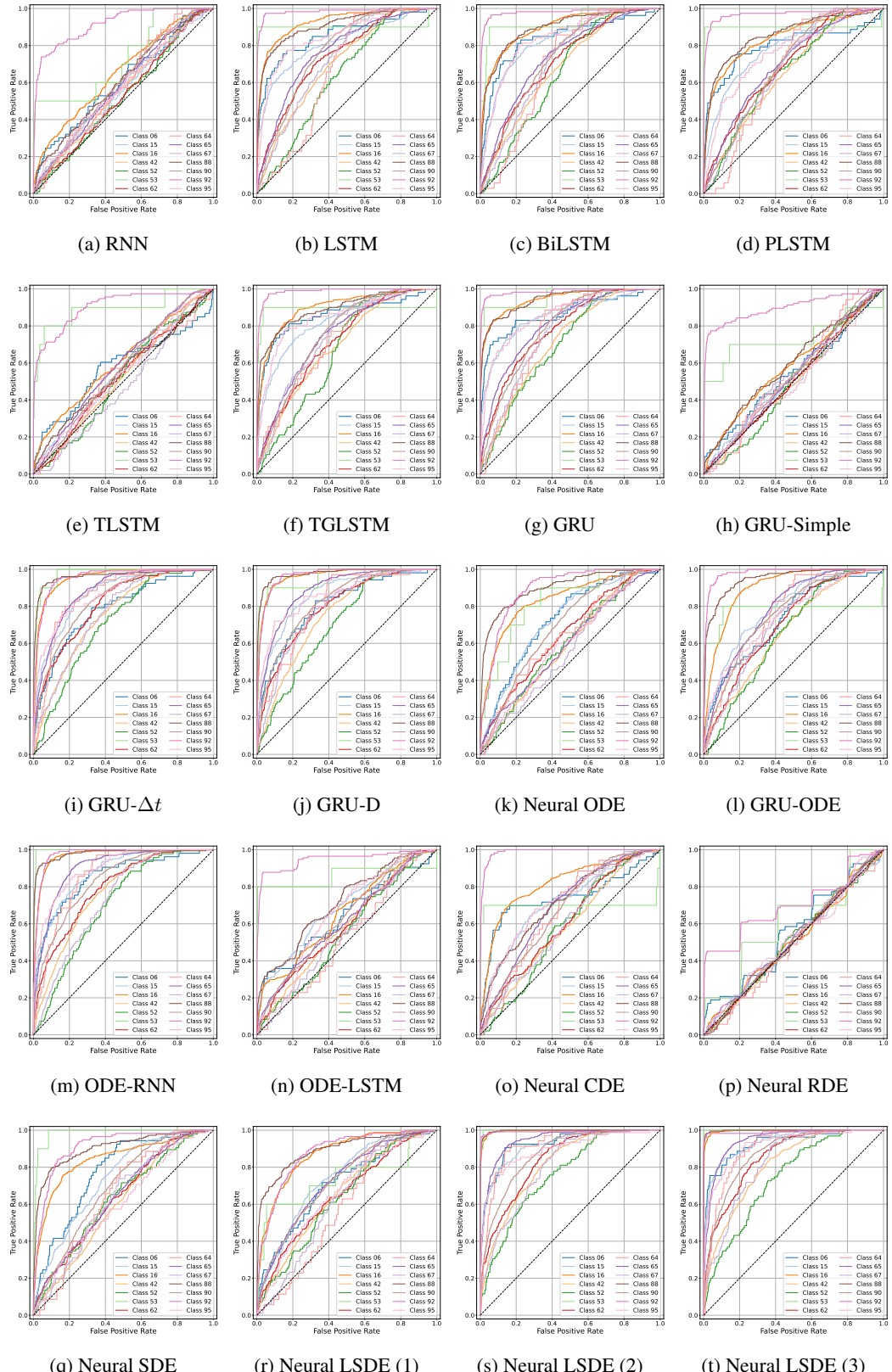

Figure 5: Receiver operating characteristic curves for each class, under the irregular scenario

