# OpenReview forum: "Neural Langevin-type Stochastic Differential Equations for Astronomical time series Classification under Irregular Observations"
_ICLR.cc/2024/Workshop/AI4DiffEqtnsInSci — AI4DiffEqtnsInSci @ ICLR 2024 Poster_

### Official Review · Reviewer_W4Bd · 2024-02-17

**Rating:** 5
**Confidence:** 3

**Review:**

**Summary of the paper:**
The authors present an analysis of three different manners to process neural stochastic differential equation initial condition for astronomical data. The initial condition is first transformed to fill in some missing data, either through cubic interpolation, mean filling, or 0 filling. This transformed condition is then passed into a latent space where the SDE is solved.

**Strength of the paper:**
The motivation of the paper are well described (the need to properly define initial conditions), and the review of neural ODEs, SDEs, and Langevin SDEs is well written. Regardless of the choice of initial condition processing technique, the use of Neural LSDE seems to outperform other neural network based approach.

**Weakness of the paper:** The three different approaches to handle initial conditions are not well described and are particularly confusing in light of Figure 1. The authors state that the goal is to handle missing data in the initial condition, but in figure 1, the missing data is filled across time. It is not clear if figure 1 represent x or z(t), and what each color represent. The author mention that in approach #2, the mean is used in-lieu of missing data, but it is not clear which mean. In figure 1.c, it rather seems like the missing data has been replace by zeros, instead of a (undefined) mean. It is also very puzzling to me that if any value is missing from x, it is fully replaced by zeros. In this case, all the initial condition with missing data would be processed into the same initial condition, and the variance across solutions would no longer be function of the initial condition, but only the SDE inherent stochasticity.

The dataset description is also unclear: Solving neural differential equations implies that we are looking at predictions that a continuous time series, but the dataset mentions classification. How do SDEs fit with classification tasks?

---

### Official Review · Reviewer_Uymz · 2024-02-24

**Rating:** 7
**Confidence:** 4

**Review:**

This paper investigates three different strategies for assigning optimal initial condition in Neural Langevin-type stochastic differential equations. The investigation focuses on the classification task involving irregular time series data in astronomical studies. The three methods for optimal initial conditions include the interpolation method, imputation method, and static approach. Through extensive experiments, this study demonstrates that Neural SDEs are robust in handling irregular time series data, and the static method for initial conditions provided the best results for their problem.
1. In the imputation method, how is the mean computed? Is it a temporal mean or the mean of input features measured at that time? From Fig. 2, it appears that missing values are mostly replaced with zeros in the imputation method. It would be helpful to mention that the black dashed-dotted line corresponds to missing data in the figure caption.
2. The classification task is not clearly defined. More details should be added about the specific problem being addressed in the LSST dataset.
3. The three strategies for optimal initial conditions are tested on only a single dataset. Are these observations dataset-specific, or is the static or imputation method consistently better than the interpolation method overall?
4. How does the computational time of Neural L-SDE compare to other methods like RNN, LSTM, and Neural SDE?

---

### Meta-Review · Area_Chair_2Ph2 · 2024-03-01

**Recommendation:** Accept (Poster)

**Metareview:**

The paper investigates three strategies for defining optimal initial conditions in Neural Langevin SDEs for time series classification. Authors find that the static initialization approach works best on astronomical data. For the camera-ready version, the authors should provide more details on the classification task, datasets, imputation techniques, justify using zero imputation and clarify the figures. I further encourage authors to discuss or implement expanding the experiments beyond one dataset, and compare computation time to other methods.

---

### Decision · Program_Chairs · 2024-03-02

Accept (Poster)